evolution, theoretical biology, health and disease and epidemiology

evolution of virulence, host heterogeneity, periodicity, treatments, vaccines, reproductive value

**Author for correspondence:**
Alicia Walter
e-mail: alicia.walter@cefe.cnrs.fr

# Epidemiological and evolutionary consequences of periodicity in treatment coverage

Alicia Walter and Sébastien Lion

CEFE, CNRS, Univ Montpellier, EPHE, IRD, Univ Paul Valéry Montpellier 3. 1919, route de Mende, Montpellier, France

AW, 0000-0002-7334-0247; SL, 0000-0002-4081-0038

Host heterogeneity is a key driver of host–pathogen dynamics. In particular, the use of treatments against infectious diseases creates variation in quality among hosts, which can have both epidemiological and evolutionary consequences. We present a general theoretical model to highlight the consequences of different imperfect treatments on pathogen prevalence and evolution. These treatments differ in their action on host and pathogen traits. In contrast with previous studies, we assume that treatment coverage can vary in time, as in seasonal or pulsed treatment strategies. We show that periodic treatment strategies can limit both disease spread and virulence evolution, depending on the type of treatment. We also introduce a new method to analytically calculate the selection gradient in periodic environments, which allows our predictions to be interpreted using the concept of reproductive value, and can be applied more generally to analyse eco-evolutionary dynamics in class-structured populations and fluctuating environments.

## 1. Introduction

Parasites are an ubiquitous threat to plant, animal and human populations. This has led to the development of numerous pre- and post-infection treatments, which play a central role in the fight against infectious diseases. At a fundamental level, this has also motivated a long line of research in epidemiology to devise control measures that can limit the potentially dramatic effects of epidemics for animal and plant species [1,2]. A key driver of this theoretical work is to find the optimal strategy to deploy treatments in order to maximize the short-term (epidemiological) benefits of treatments while mitigating their potential long-term (evolutionary) negative consequences.

Indeed, it is increasingly acknowledged that pathogens may evolve in response to treatments, as exemplified by the evolution of antibiotic resistance or vaccine escape. Pathogen evolution is fuelled by the high reproduction rate of pathogens and the increasing use of treatments. In particular, both empirical evidence [3–5] and theoretical predictions [6–8] support the idea that imperfect treatments may cause selection on pathogen life-history traits, such as transmission and virulence.

In practice, treatments are rarely perfect, either because of partial efficacy, or limited coverage. From an ecological perspective, this introduces heterogeneity in the host population. Indeed, naive and treated hosts can be viewed as two habitats with different qualities for the pathogen. We expect pathogens to have higher fitness in good habitats (e.g. untreated hosts), and lower fitness in bad habitats (e.g. treated hosts). However, if the pathogen adapts to the bad habitat by increasing its virulence, this can negatively impact untreated hosts in the population. The potential negative effects of such imperfect treatments have been theoretically studied, showing in particular that some types of treatments (notably those reducing host susceptibility or pathogen transmissibility) can be viewed as evolution-proof, while others (such as those that

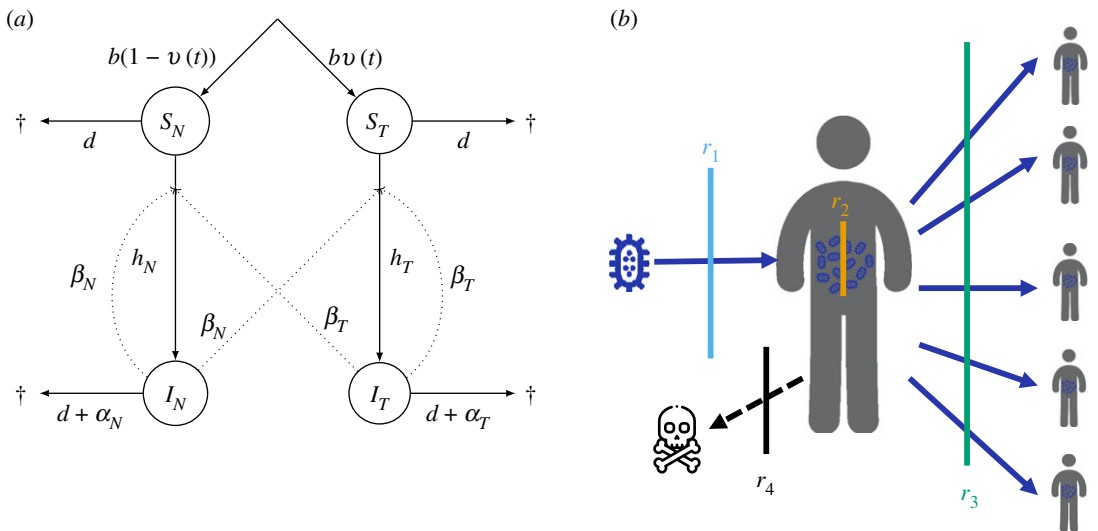

**Figure 1.** (a) Life cycle of the host–pathogen interaction, showing transition rates between classes. (b) Infection process and the different treatment mechanisms: anti-infection ($r_1$, blue), anti-growth ($r_2$, orange), anti-transmission ($r_3$, green) and anti-toxin ($r_4$, black), with a human as illustration of hosts. (Online version in colour.)

mainly increase the tolerance of hosts to the disease) can lead to the long-term evolution of virulence [7–9].

A central assumption of these earlier studies is that treatment coverage is constant in time. In practice, however, treatment coverage fluctuates in time, either because of social or economical constraints (vaccine scares, shortage in drugs, vaccination campaigns linked to time-limited humanitarian or scientific missions), or because of specific treatment strategies. For instance, pulse vaccination has been the main strategy deployed to eradicate polio and significantly decreased its incidence [10]. In addition, many crop diseases are seasonally treated, either directly in fields, or by pre-treatment at the seed stage. Periodic treatments may also be implemented through rotation between naive and treated plants, which has been shown to reduce pathogen prevalence [11]. In this context, it is of major interest to study the consequences of periodic treatment strategies through theoretical studies to guide future experimentations or applications. In particular, an important question is whether temporal variations in treatment may increase or decrease selection on pathogen virulence or transmission. At a conceptual level, fluctuations in treatment coverage will cause temporal variation in host quality for the pathogen, and it is not clear that the optimal strategy in a constant environment will also maximize pathogen fitness in this context.

Our purpose here is to analyse the epidemiological and evolutionary effects of periodic imperfect prophylactic treatments that create heterogeneity among hosts. Building on previous theoretical studies of imperfect treatments with constant coverage [7–9], we first present the consequences of periodic treatment coverage on the prevalence of the disease and the pathogen's basic reproduction number. We then assume that a rare mutation occurs in the pathogen, modifying the life-history traits, and we analyse how the selective pressures on the mutant pathogen depend on (i) the mode of action, and (ii) the temporal distribution of treatments. Part of our analysis is based on a new approach to analyse selection in fluctuating environments, which allows us to measure host quality using a dynamical concept of reproductive value [12]. We discuss the practical and conceptual implications of our work.

## 2. Model

We consider a host–pathogen interaction with the life cycle depicted in figure 1a (see table 1 for notations). The host population is structured in two classes corresponding to the host's immune state, naive (N) or treated (T). Hosts can be either susceptible (S) or infected by a pathogen responsible for an infectious disease (I). This leads to four different types of hosts, $S_N$, $S_T$, $I_N$ and $I_T$. Host reproduction occurs at rate b. Upon birth, offspring have a probability v of being treated, in which case they enter the $S_T$ class, and a probability $(1 - v)$ of remaining untreated, in which case they enter the $S_N$ class. All hosts (susceptible and infected) have a background mortality rate, d, with an additional disease-induced mortality (i.e. virulence), which we note $\alpha_N$ and $\alpha_T$, respectively, for naive or treated infected hosts. A susceptible naive (respectively, treated) host can be infected by naive and treated infected hosts, $I_N$ and $I_T$, at rate $h_N$ (respectively, $h_T$). The forces of infections, $h_N$ and $h_T$, directly depend on the horizontal transmission of the pathogen, such that $h_N = \beta_N I_N + \beta_T I_T$ and $h_T = \sigma h_N$, where $\beta_N$ and $\beta_T$ are the transmissibilities of naive and treated hosts, respectively, and $\sigma$ represents the relative susceptibility of treated hosts. With these assumptions, the epidemiological dynamics can be described by the following ODE system

$$\frac{dS_N}{dt} = (1 - v(t))b - (d + h_N)S_N, \tag{2.1a}$$

$$\frac{dS_T}{dt} = v(t)b - (d + h_T)S_T, \tag{2.1b}$$

$$\frac{dI_N}{dt} = h_N S_N - (d + \alpha_N)I_N \tag{2.1c}$$

and $$\frac{dI_T}{dt} = h_T S_T - (d + \alpha_T)I_T, \tag{2.1d}$$

The model is based on a previously analysed model of imperfect vaccines [7,8], but instead of a constant treatment coverage, we consider that v is a function of time. In most simulations, we use a T-periodic square function (figure 2a) that varies between $v_{min}$ and $v_{max}$. The treatment coverage takes the value $v_{max}$ during $pT$ and $v_{min}$ during $(1 - p)T$, with p the fraction of the period with a maximum coverage.

**Table 1.** Table of parameters and variables.

| parameter | definition |
| --- | --- |
| **life-history traits** | |
| $\alpha_k$ | disease-induced death rate (virulence) of the resident strain in class $k$ |
| $\alpha_k'$ | disease-induced death rate (virulence) of the mutant strain in class $k$ |
| $\alpha^*$ | evolutionarily stable virulence |
| $\beta_k$ | transmission rate of the resident strain in class $k$ |
| $\beta_k'$ | transmission rate of the mutant strain in class $k$ |
| $b$ | birth rate |
| $d$ | natural death rate |
| $h_k$ | force of infection in class $k$ |
| $\sigma$ | susceptibility of hosts |
| **treatments** | |
| $v(t)$ | periodic treatment coverage function |
| $T$ | period of the treatment function |
| $p$ | fraction of T with treatment |
| $r_i$ | efficacy of the $i$ treatment (see §2) |
| **reproductive values** | |
| $c_k^e$ | pathogen class reproductive value in class $k$ under a constant coverage |
| $c_k$ | pathogen class reproductive value in class $k$ under a periodic coverage |
| | $k$ stands for naive ($N$) or treated ($T$) hosts |

The average coverage is thus $\bar{v} = p\,v_{\max} + (1-p)\,v_{\min}$. For the sake of simplicity, we choose $v_{\max} = 1$ and $v_{\min} = 0$ in our simulations, so that the average coverage is $\bar{v} = p$.

As typical in the literature, we assume a trade-off between transmission and virulence, so that a pathogen cannot see its transmission increase without paying a cost through an increase of the host's death [13,14]. Following this hypothesis, we assume an increasing saturating trade-off between transmission, $\beta$ and virulence, $\alpha$. In figures, we typically use the function $\beta[\alpha] = 5\alpha/(1+\alpha)$.

As in [7–9], we assume that treatments are imperfect: being treated does not guarantee a full and lifelong protection against pathogens. Several imperfect vaccines against cholera for instance have been reported, with various efficacies [15,16]. Following [7], we consider four different types of treatments showed in figure 1b. First, we consider an anti-infection treatment that prevents the primary infection of the host by the pathogen. Potential examples include the human papillomavirus (HPV) vaccine, which aims to reduce the penetration of the HP virus into cells [17], or the copper in Bordeaux mixture, which lowers the risk of infection by preventing the germination of fungal spores on leaves [18]. Second, we consider an anti-growth treatment which aims to decrease the intra-host growth rate of the pathogen, and has an action on its life-history traits (virulence $\alpha$ and transmission $\beta$). This is reminiscent of the mode of action of the ABT-538 drug, which reduces the within-host replication of HIV [19,20]. The third treatment acts by reducing the transmission of pathogens to other hosts. For instance, it has been shown that an effect of the feline calcivirus vaccine is to

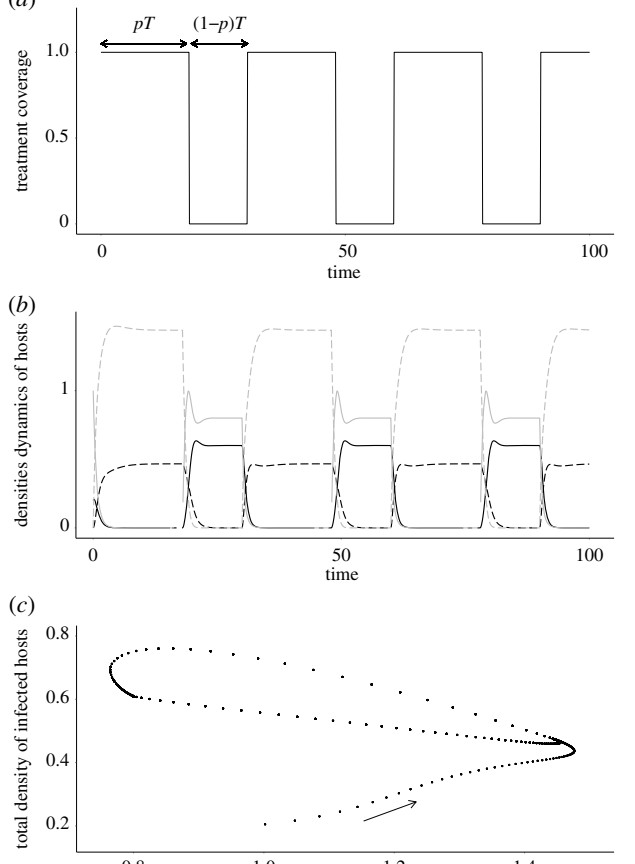

**Figure 2.** Typical behaviour of the model, for an anti-growth treatment. (a) Square function for treatment coverage: $v$ takes the value 1 during $pT$ (all newborns are treated), and the value 0 during $(1-p)T$ (no newborn is treated). (b) Density dynamics for susceptible naive (plain grey), susceptible treated (plain black), infected naive (dashed grey) and infected treated (dashed black). (c) Phase portrait for the total densities of susceptible ($S_N + S_T$) and infected ($I_N + I_T$) hosts, showing convergence to a periodic attractor. Parameter values: $r_2 = 0.8$, $p = 0.6$, $T = 30$, $\alpha = 1$, $b = 2$, $d = 1$. Typical behaviours of the model with anti-infection, anti-transmission and anti-toxin treatments are shown in electronic supplementary material, appendix S.1.

reduce virus shedding [21,22]. Fourth, we consider a leaky anti-toxin treatment that only reduces pathogen's virulence, as documented for instance for the vaccine against Marek's disease [4], or the toxoid vaccine against diphtheria [3].

In naive hosts, the virulence and transmission rates are simply $\alpha_N = \alpha$ and $\beta_N = \beta(\alpha)$, where $\alpha$ is the virulence trait. Treatments cause a reduction in virulence, transmission or susceptibility of hosts, depending on the treatment type $i$, with an efficacy $r_i$ that takes values between 0 (no effect) and 1 (perfect treatment). With our definitions for the different types of treatments, we have

$$\sigma = 1 - r_1, \quad \alpha_T = (1 - r_2)(1 - r_4)\alpha, \quad \beta_T = (1 - r_3)\beta[(1 - r_2)\alpha]. \tag{2.2}$$

## 3. Epidemiology

In this section, we investigate how the periodicity of treatment coverage affects the ability of a pathogen to spread in an uninfected host population and the endemic prevalence of the disease. The invasion success of a pathogen can be quantified by its basic reproduction number, $R_0$, which represents the average number of hosts to which a single infected host in a disease-free population can transmit the

pathogen over its lifetime. In a deterministic model, the pathogen can create an epidemic if $R_0 > 1$, or die out if $R_0 < 1$ [23–25]. $R_0$ is calculated in the stable population in the absence of the disease, typically an equilibrium in most studies. In our model, however, the disease-free population settles on a periodic attractor, due to the periodic forcing $v(t)$.

## (a) Constant treatment coverage

When the treatment coverage is constant ($v(t) = \bar{v}$), the population settles on an equilibrium in the absence of the disease. We can therefore calculate $R_0$ using the next-generation theorem (see electronic supplementary material, appendix S.2.1, [24,26]), which leads to

$$R_0 = R_N S_N^0 + \sigma R_T S_T^0 \tag{3.1}$$

with

$$R_N = \frac{\beta_N}{d + \alpha_N} \quad \text{and} \quad R_T = \frac{\beta_T}{d + \alpha_T}, \tag{3.2}$$

the basic reproductive numbers in a fully naive ($\bar{v} = 0$) or fully treated ($\bar{v} = 1$) population respectively, and $S_N^0$ and $S_T^0$ the densities of susceptible hosts at the disease-free equilibrium, which are given by $S_N^0 = (1 - \bar{v})b/d$ and $S_T^0 = \bar{v}b/d$ (electronic supplementary material, appendix S.3). Thus, the basic reproductive number in a heterogeneous population corresponds to the sum of $R_0$'s in a fully naive and a fully treated population, weighted by the densities of susceptible hosts in each class.

## (b) Periodic treatment coverage

When host dynamics are periodic (figure 2a,b), the next-generation theorem cannot be applied to calculate the basic reproductive number. However, the concept of Floquet multipliers can be used to extend the definition of $R_0$ [24] to periodic environments [27,28]. Floquet theory allows for the study of the stability of continuous-time periodical systems ([29]; e.g. [30] for an ecologically oriented treatment). In our study, we can distinguish two cases: anti-infection and anti-transmission treatments, for which an analytical study is possible, and anti-growth and anti-toxin treatments for which $R_0$ can only be calculated numerically.

### (i) Anti-infection and anti-transmission treatments

For these treatments, we show in electronic supplementary material, appendix S.2.2 that, following [27], we can calculate the basic reproduction number as:

$$R_0 = \frac{\beta}{d + \alpha} \left( \langle S_N^0 \rangle + (1 - r_1)(1 - r_3)\langle S_T^0 \rangle \right), \tag{3.3}$$

where $\langle S_k^0 \rangle = \int_t^{t+T} S_k^0(\tau)\,d\tau$ gives the average density of susceptible hosts in class $k$ during one period. In particular, we show in electronic supplementary material, appendix S.3 that, for our model

$$\langle S_N^0 \rangle = \frac{(1 - \bar{v})b}{d} \quad \text{and} \quad \langle S_T^0 \rangle = \frac{\bar{v}b}{d}. \tag{3.4}$$

We can see that the average susceptible densities, $\langle S_N^0 \rangle$ and $\langle S_T^0 \rangle$ have the same expressions as the equilibrium densities in the constant case, and only depend on $b, d$, and the average treatment coverage, $\bar{v}$ (equation (3.4)). As a result, the invasion threshold is independent of the period, and only depends on the average coverage $\bar{v}$. Hence, any periodic treatment with

average coverage $\bar{v}$ leads to the same condition for pathogen invasion as a constant treatment with coverage $\bar{v}$.

After pathogen invasion, the population settles on a periodic endemic attractor (figure 2c), and the average prevalence is then the fraction of infected hosts in the population integrated over one period. Increasing the average coverage ($\bar{v} = p$) decreases the average prevalence for these treatments (figure 3a, blue and green lines), until extinction above a critical value of $p$. Moreover, an increase of the period $T$ also reduces the average prevalence (figure 3b). Note that periodicity in treatment coverage always leads to a lower prevalence compared to the corresponding constant coverage.

### (ii) Anti-growth and anti-toxin treatments

For these treatments, the basic reproduction number cannot be expressed in a closed analytical form, but can be calculated numerically using Floquet's theory [30,31]. We show in electronic supplementary material, appendix S.2.3 that the effect of periodicity on $R_0$ is weak, but tends to slightly lower the extinction threshold for both types of treatments.

As the average coverage ($\bar{v} = p$) increases, the average prevalence decreases with an anti-growth treatment and increases with an anti-toxin. This can be intuitively explained by the fact that anti-toxin treatments merely reduce the survival cost of pathogens and cause infected hosts to transmit for a longer time. By contrast, anti-growth treatment directly impacts the transmission-virulence trade-off. It causes a reduction of pathogen spread for treated hosts, counterbalanced by a reduction of host mortality. This explains the lower decrease of the prevalence compared to anti-infection or anti-transmission treatments (figure 3a). As for previous treatments, the average prevalence decreases when the period increases and the periodic treatment is beneficial compared to the constant coverage in terms of average prevalence reduction (figure 3b).

To understand why, note that the prevalence for short periods can be approximated by the prevalence with the corresponding constant average coverage. Indeed, the dynamics are characterized by small oscillations around the equilibrium solution. Hence, for short periods, pathogens are exposed, at any given time, to a heterogeneous population of naive and treated hosts. For large periods, however, the environment experienced by the pathogen alternates between temporary equilibria corresponding to either fully naive or fully treated populations. The average prevalence then tends to the arithmetic mean between the equilibrium prevalences in fully naive and fully treated poulations, which is always lower than the prevalence in a heterogeneous population at equilibrium (figure 3b). Note that, due to the epidemiological feedbacks, the endemic attractor is more sensitive to the forcing period than the disease-free state of the population, which explains why periodicity has a weaker impact on the invasion threshold compared to the average prevalence.

## 4. Evolution

Assuming the population has reached an endemic attractor, we now aim to understand how imperfect treatments and periodicity may jointly affect the evolution of pathogen life-history traits, such as virulence. In this section, we consider that a pathogen strain is characterized by a trait $z$ (for instance the pathogen's within-host growth rate) and that the virulence and transmission traits are all functions of this trait. For

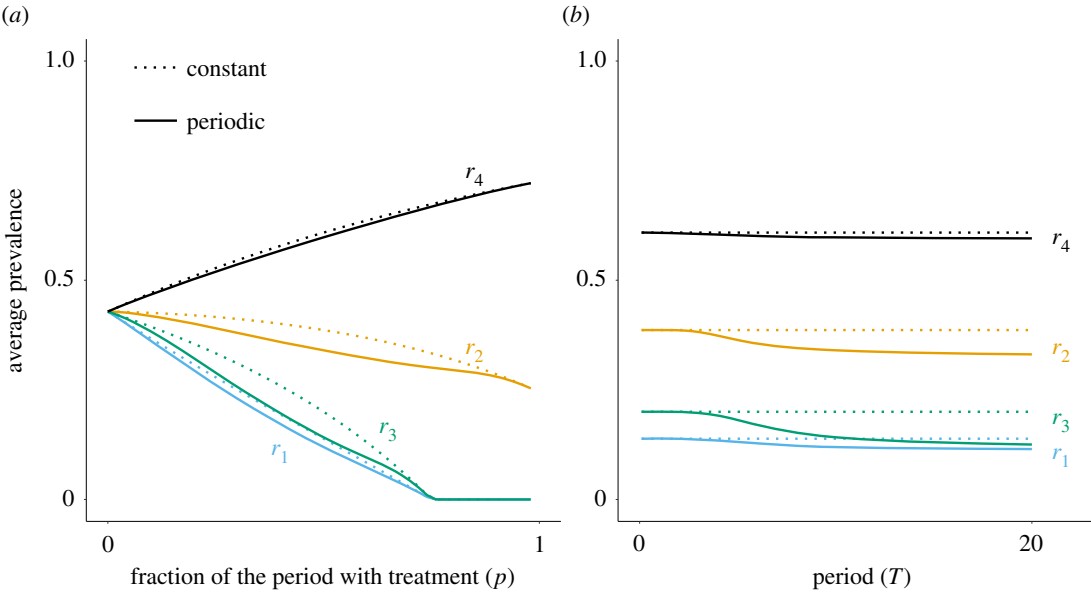

**Figure 3.** Disease prevalence as a function of (*a*) the fraction of time at maximum treatment $p$ ($T = 10$), or (*b*) the period $T$ ($p = 0.5$), for anti-infection (blue), anti-growth (orange), anti-transmission (green) and anti-toxin (black) treatments, all with efficacy $r_i = 0.8$. (Solid lines: periodic treatment, dotted lines: constant treatment.) Other parameters as in figure 2. (Online version in colour.)

simplicity, we assume $\alpha_N = z$, so that $z$ represents virulence in naive hosts. Eventually, a host population infected by the resident strain settles on its endemic attractor ($\hat{S}_N(t)$, $\hat{S}_T(t)$, $\hat{I}_N(t)$, $\hat{I}_T(t)$). Extensive numerical simulations show that the period of the attractor is always equal to the period of $v(t)$. Now let us assume that a mutant with a trait $z'$ which is slightly different from the resident's appears (the full model including the infected hosts by the mutant strain is shown in electronic supplementary material, appendix S.4.1). Whether the mutant can invade the population or not is determined by the sign of its invasion fitness [32–35]. We use this conceptual tool to investigate the direction of selection on pathogen traits, and potential evolutionary endpoints.

## (a) Anti-infection and anti-transmission treatments

For these treatments, the invasion fitness of a rare mutant pathogen can be calculated analytically from the mutant pathogen's basic reproduction number $R(z', z)$, which acts as a proxy for invasion fitness [25]. Using the same approach used to derive equation (3.3), we find that a rare mutant can invade the resident population on its periodic attractor if $R(z', z) > 1$, where

$$R(z', z) = \frac{\beta(z')}{d + \alpha(z')}\left(\left\langle \hat{S}_N \right\rangle + (1 - r_1)(1 - r_3)\left\langle \hat{S}_T \right\rangle\right). \quad (4.1)$$

Because for the resident pathogen, we have $\left\langle \hat{S}_N \right\rangle + (1 - r_1)(1 - r_3)\left\langle \hat{S}_T \right\rangle = (d + \alpha(z))/\beta(z) = 1/R_0(z)$, it follows that

$$R(z', z) = \frac{R_0(z')}{R_0(z)}. \quad (4.2)$$

Equation (4.2) shows that selection favours the strain that maximizes the epidemiological $R_0$, as in the case with constant coverage [7–9]. As a result, periodicity in treatment coverage does not impact the evolutionarily stable virulence for these two treatments. This can be numerically confirmed using Floquet's theory (electronic supplementary material, figure S.5). For completeness, we note that, in the original models by Gandon *et al.* [7,8], the prediction that higher efficacy selects for lower virulence was due to the possibility of

superinfection, which we ignore here, as a full analysis of the interplay between multiple infections and environmental fluctuations is beyond the scope of this paper.

## (b) Anti-growth and anti-toxin treatments

For anti-growth and anti-toxin treatments, an analytical expression of the invasion fitness $s(z', z)$ (or its proxy $R(z', z)$) is beyond our reach. However, we use a new approach to derive an analytical expression of the selection gradient, $\mathcal{S}(z) = \partial s/\partial z'|_{z'=z}$. The zeros of the selection gradient give the evolutionarily singular points [36].

As a useful baseline scenario, we first examine the case of a constant coverage analysed in [7,8]. We then have

$$\mathcal{S}_e = c_N^e\left(\beta_N'(z)\frac{d + \alpha_N}{\beta_N} - \alpha_N'(z)\right) + c_T^e\left(\beta_T'(z)\frac{d + \alpha_T}{\beta_T} - \alpha_T'(z)\right). \quad (4.3)$$

The selection gradient then takes the form of a weighted sum, where the weights $c_N^e$ and $c_T^e = 1 - c_N^e$ are the pathogen's class reproductive values in naive and treated hosts, in a resident population at equilibrium [9,12,37,38]. Thus, the ES virulence is intermediate between the value which maximizes the basic reproduction number in a fully naive population (i.e. when $c_N^e = 1$) and the value which that maximizes the basic reproduction number in a fully treated population (i.e. when $c_T^e = 1$). The class reproductive value $c_T^e$ is a measure of the quality of treated hosts from the point of view of the parasite.

With periodic treatment coverage, we show in electronic supplementary material, appendix S.4.2 that the average change in mean trait over one period on the resident periodic attractor, is approximately proportional to

$$\mathcal{S} = \langle c_N \rangle\left(\beta_N'(z)\frac{d + \alpha_N}{\beta_N} - \alpha_N'(z)\right) + \langle c_T \rangle\left(\beta_T'(z)\frac{d + \alpha_T}{\beta_T} - \alpha_T'(z)\right), \quad (4.4)$$

where $c_T(t)$ is the class reproductive value of resident pathogens infecting treated hosts at time $t$, and $c_N(t) = 1 - c_T(t)$ is the class reproductive value on naive hosts. Note that equation (4.4) is directly comparable to equation (4.3), but, in contrast with the classical definition, reproductive values are here time-dependent quantities [12]. This reflects the fact that the relative qualities of the different classes of hosts for the pathogen change as the population moves along the periodic attractor.

Equation (4.4) shows that, as in the constant treatment model, the ES virulence in a heterogeneous host population is intermediate between the value which maximizes the basic reproduction number in a fully naive population and that maximizes the basic reproduction number in a fully treated population. Interestingly, the potentially complex effects of fluctuations on the exact value of the ESS are captured by a single variable, which is the average class reproductive value over one period, $\langle c_T \rangle$. In general, $\langle c_T \rangle$ will deviate from the class reproductive value $c_T^e$ in the constant treatment model (figure 4$a$–$c$), and as a result the ESS will be different in periodic versus constant environments.

For anti-toxin treatments, where $\beta_N = \beta_T = \beta(\alpha)$ and $\alpha_T = \alpha(1 - r_4)$, a useful graphical representation can be obtained from equation (4.4) by noting that the ES virulence must satisfy

$$\beta'(\alpha) = \frac{\beta(\alpha)}{d + \alpha} \frac{1 - r_4 \langle c_T \rangle}{1 - r_4 \langle c_T \rangle \frac{\alpha}{d + \alpha}} \tag{4.5}$$

which is a form of marginal value theorem [39]. As shown graphically in figure 4$d$, the ES virulence for periodic anti-toxin treatment is lower than in a constant treatment. This effect is mediated by the average class reproductive value which decreases as the period $T$ increases (figure 4$c$). For large $T$, $\langle c_T \rangle$ converges towards $\bar{\nu} = p$ (electronic supplementary material, appendix S.4.2), which provides a lower bound for the reduction in virulence that can be achieved by using periodic treatments. For our trade-off function $\beta(\alpha) = 5\alpha/(1 + \alpha)$, this lower bound can be calculated as

$$\alpha^* = \frac{1}{\sqrt{1 - pr_4}}, \tag{4.6}$$

which implies that, for a fixed efficacy $r_4$, the ES virulence should be lower for low values of $p$ (e.g. short bouts of treatment).

While the average value $\langle c_T \rangle$ is sufficient to determine the ES virulence, the impact of periodicity on the full dynamics of $c_T(t)$ sheds light on the process by which higher periods select for lower virulence. Figure 4$a$ shows that, for short periods, the class reproductive value rapidly fluctuates around its mean value, which is close to the equilibrium value $c_T^e$ but overestimates the average state of the environment, $\bar{\nu}$. As the period increases, the environmental change becomes slower and easier to track by the pathogen, and the class reproductive value more closely matches the environmental signal $\nu(t)$ (figure 4$b$). Because the pathogen is now better able to perceive the true alternance of good and bad epochs, its optimal strategy is less biased towards treated hosts. Note that equation (4.4) also allows us to understand why anti-infection and anti-transmission treatments are insensitive to periodicity, since, for these treatments, the class-specific selection gradients (the terms between brackets) are equal, so that both classes have the same optimum.

These analytical predictions can be checked using a numerical calculation of the invasion fitness $s(z', z)$, which is the Floquet exponent associated with the mutant dynamics on the resident periodic attractor (see electronic supplementary material, appendix S.4.3). Figure 5$a$ shows that the predictions of equation (4.4) closely match the Floquet analysis and confirms that the ES virulence decreases as the period increases, with a stronger effect for anti-growth than for anti-toxin treatments. There is however a lower bound to the reduction in virulence that can be achieved using periodic treatments, as the ES virulence saturates as $T$ becomes large. Nonetheless, a tentative conclusion of our work is that selection for virulence is weaker with treatments with periodic coverage such as pulse vaccination, compared to a constant treatment with similar average coverage. Finally, figure 5$b$ shows that, for both anti-growth and anti-toxin treatments, increasing $p$, and thus the average treatment coverage in the population, leads to increased ES virulence, as in the constant case. However, in heterogeneous host populations $(0 < p < 1)$, the ES virulence is lower with periodic treatments than with constant treatments. Hence, although, as predicted by Gandon *et al.* [7,8], anti-growth and anti-toxin treatments select for higher virulence, fluctuations in coverage may actually mitigate the negative evolutionary side-effect of these treatments.

## 5. Discussion

Our study sheds light on the potential benefits of periodic treatments, both for short-term epidemiological control of infectious diseases and long-term virulence management. Our model generalizes the predictions of [7,8], who showed for constant treatment coverage that increasing coverage decreases the endemic prevalence, except for anti-toxin treatments. Our results show that this carries over to periodic coverage. For long-term evolution, we show that increasing average treatment coverage has no effect on virulence evolution for anti-infection and anti-transmission treatments but selects for significantly increased virulence with anti-growth and anti-toxin treatments. However, for all treatments, periodic treatments are more beneficial than constant treatments and lead to lower prevalence and virulence.

Our results suggests that periodic treatments over long periods may be a mitigating strategy for virulence management, even for treatments that create selective pressures on pathogen life-history traits. Overall, our model predicts that the best treatment strategy with anti-infection and anti-transmission is a periodic coverage with a high average coverage, to lower the prevalence. Anti-growth treatment strategies require a compromise between high coverage (reduction of prevalence on the short term) and low coverage (to limit the long-term emergence of virulent strains). Anti-toxin treatments are not recommended, but if no alternative exists, periodic strategies of treatment would seem preferable.

At a conceptual level, our modelling approach provides a general method to analyse selective pressures in heterogeneous habitats where habitat quality can vary over time due to environmental fluctuations. Here, the quality of treated hosts for the parasite varies, and we use a dynamical concept of reproductive value [12] to measure this quality and derive the selection gradient. In contrast to previous studies [40,41], which relied on numerical calculations of the invasion fitness, this allows us to derive analytical expressions that can be

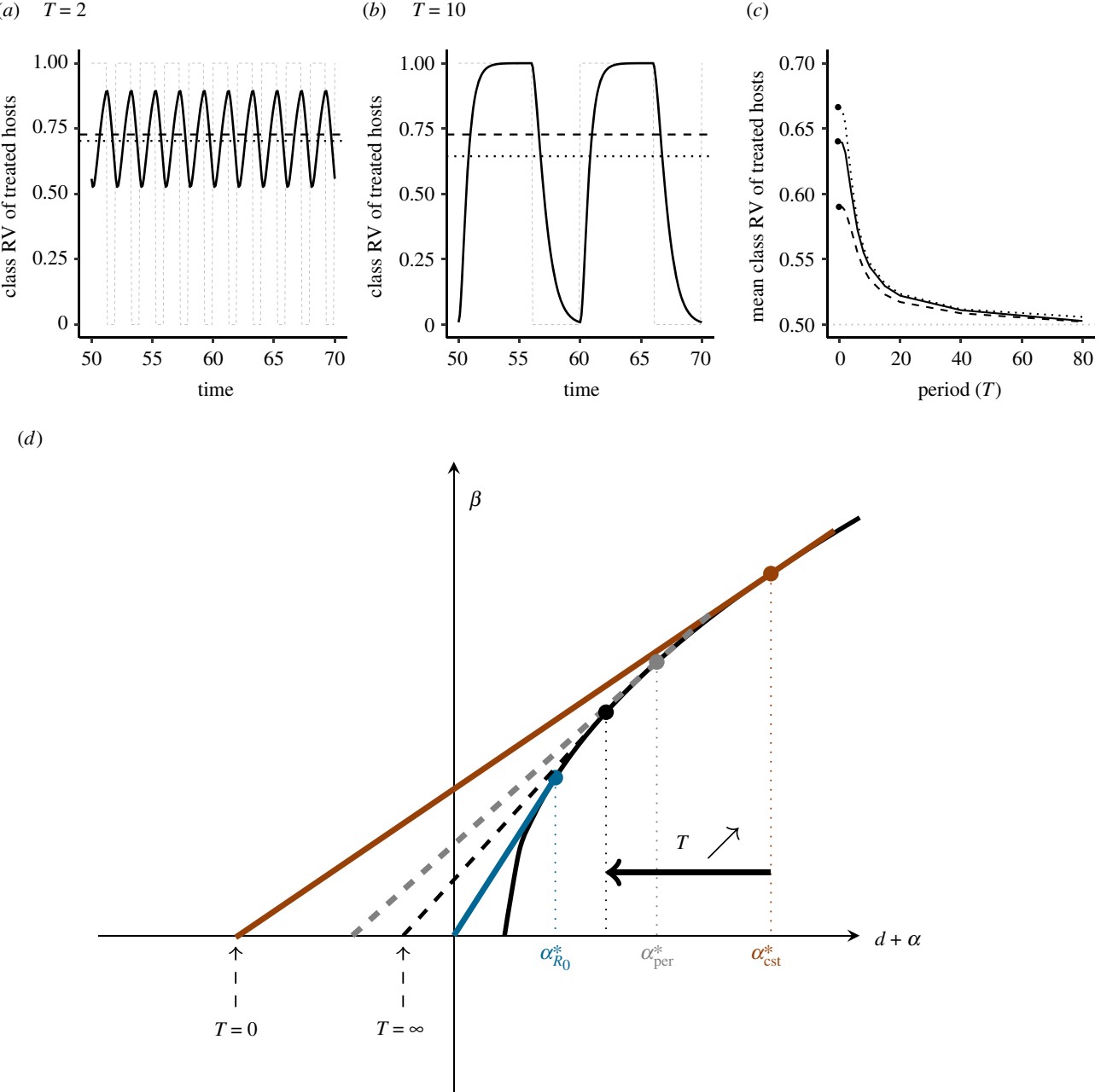

**Figure 4.** (a,b) Dynamics of the class reproductive value of treated hosts, $c_T(t)$ for short (a, $T = 2$) and longer (b, $T = 10$) period of an anti-toxin treatment coverage. Also shown are the periodic treatment function $v(t)$ (dashed grey lines), the mean value $\langle c_T(t) \rangle$ (dotted lines) and the value in the constant model $c_T^e$ (dashed line). Parameters: $r_4 = 0.5$, $p = 0.6$, $z = 1$. (c) Mean class reproductive value $\langle c_T(t) \rangle$ as a function of the period. Parameters: $r_4 = 0.5$, $\bar{v} = p = 0.5$, $z = 0.5$ (dashed), 1 (solid), 1.4 (dotted). Other parameters as in figure 2. (d) Graphical representation of the effect of periodicity on the ES virulence for an anti-toxin treatment. In a homogeneous population, the ES virulence $\alpha_{R_0}^*$ is obtained by maximizing the epidemiological $R_0$. Graphically, this implies that the slope at the ESS goes through the origin (blue line). For a constant treatment ($T = 0$), the slope at the ESS is smaller (red line), resulting in a higher ES virulence $\alpha_{cst}^*$. As the period increases, so does the slope at the ESS, given by equation (4.5), resulting in a lower ES virulence. The slope at the ES virulence (grey line) then goes through the point $(-d\, r_4\langle c_T\rangle)/(1 - r_4\langle c_T\rangle), 0)$, where $\langle c_T \rangle$ is the average class reproductive value of treated hosts over one period. Because, as the period $T$ goes to infinity, $\langle c_T \rangle$ converges to its lower bound $p$ (figure 4c), the ES virulence under periodic anti-growth treatment coverage also has a lower bound, which is always greater than $\alpha_{R_0}^*$ (which would be the solution for $\langle c_T \rangle = 0$). (a) $T = 2$, (b) $T = 10$. (Online version in colour.)

readily compared to the selection gradient in constant environments. This is particularly useful as it allows us to capture the effect of periodic coverage on virulence evolution through a single variable, which is the average quality (or reproductive value) of treated hosts over one period.

In our study, we focus on periodicity caused by the availability of treatments, which implies seasonality in the susceptible host type and impacts pathogens life-history traits. However, periodic environmental forcing can be caused by other parameters such as seasonality in pathogen transmission

rate [42] or in host birth rates [43], to which our approach could be applied. In particular, considering the effect of seasonal variations of the environment caused by climate change on the evolution of pathogens life-history traits is of particular interest [44]. Many epidemiological studies have addressed the issue of periodic environments in epidemiology, in particular on the expression of $R_0$ and the probability of emergence [27,28,31,42,45]. There is however a lack of studies about evolution in fluctuating environments, notably when the population is structured. Partly, this is due to the lack of

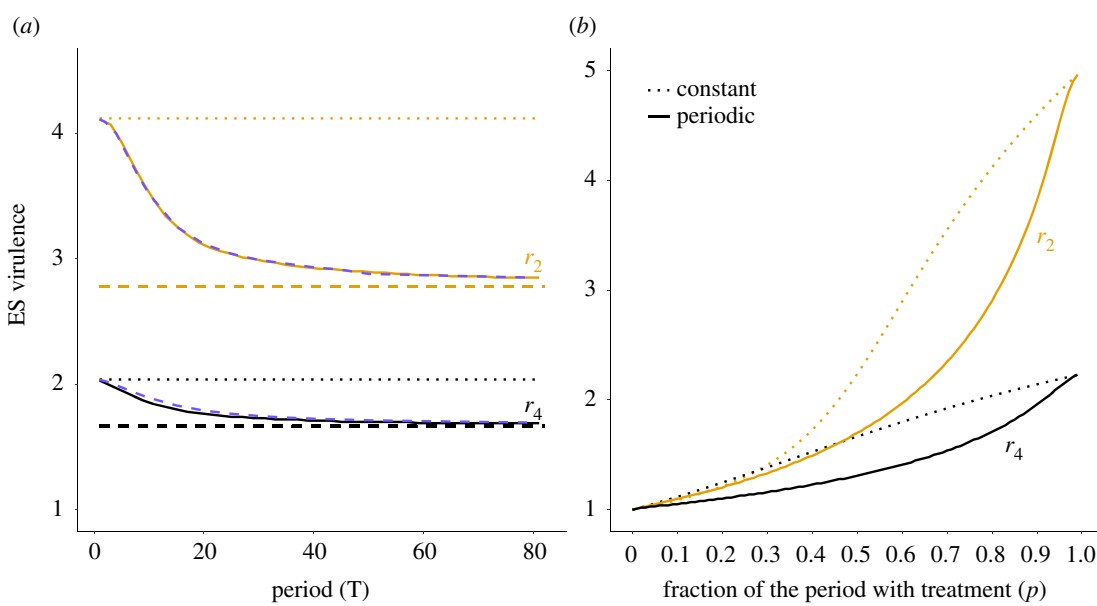

**Figure 5.** Evolutionarily stable virulence according to (*a*) the period of treatments fluctuations *T* (with $p = 0.8$), (*b*) the fraction of time at maximum treatment *p* (with $T = 50$), for anti-growth (orange) and anti-toxin (black) treatments. Dotted lines show the ESS calculated with a constant average coverage, solid lines show the ESS calculated using the dominant Floquet exponent, compared to the semi-analytical solution of equation (4.4) (dashed purple lines in (*a*)). The ES virulence is bounded by a lower bound obtained by replacing $\langle c_T \rangle$ with *p* in equation (4.4) (dashed lines). All with a treatment efficacy of $r_i = 0.8$, and other parameters as in figure 2. (Online version in colour.)

analytical methods to tackle this question. For instance, Ferris & Best [40,41] have analysed the evolution of host defence in fluctuating environments, but had to resort to numerical calculations of Floquet exponents when selection depends on both susceptible and infected host classes (e.g. when recovery is not negligible). This makes a direct comparison with constant environments difficult. By contrast, the approach we use in this paper allows us to derive an insightful analytical expression for the selection gradient and to capture much of the complexity of environmental fluctuations using the biologically meaningful concept of reproductive value. We think this approach can be more generally applied to analyse other evolutionary scenarios.

In this work, we focus on the evolution of life-history traits and do not consider the possibility that pathogens evolve resistance to treatments. We have shown that increasing the period of treatment leads to a decrease both in the prevalence and the evolutionarily stable virulence, and seems to be a good treatment strategy. However, large periods means that pathogens are potentially exposed to a treated population during a long time. In principle, this could favour the evolution of pathogen resistance, so that too long a period would probably also have unwanted effects. To sharpen these predictions, our model could be extended to consider other treatements strategies which aim to reduce pathogen resistance, such as combination therapies (patients are treated with several drugs at the same time), mixing (different patients are treated with different drugs) or cycling (each patient is given different drugs in alternance) [46]. For instance, it has been shown that combination therapies are more beneficial than mixing strategies, which are in turn more beneficial than cycling strategies [47]. It would be interesting to couple these models with fluctuations in coverage to investigate the robustness of these public health strategies.

There are a number of interesting potential extensions to our study. For instance, the treatment coverage could be a function of disease prevalence, so that the treatment strategy would vary with the spread of the disease. Also, our model assumes that only new individuals (either by birth or migration) are treated, but for some treatments a global coverage over the whole population would be more realistic. Third, it could be interesting to investigate what happens when treated hosts lose their immunity after some time and join the respective susceptible or infected untreated class. Williams & Kamel [48] have considered a similar heterogeneous model where hosts can switch class over their lifetime. The authors have shown that if an infected host transits from a class with a high reproductive value to a class with a lower reproductive value, selection favours increased host exploitation and therefore increased virulence. These results are consistent with ours, which suggests that the loss of immunity over time would not significantly impact our conclusions.

Our main results were obtained under a number of assumptions. First, we assumed no recovery of infected hosts. This hypothesis is relaxed in electronic supplementary material, appendix S.5 where we explore the effect of non-zero recovery rates. We show that recovery reduces prevalence regardless of the treatment, and decreases the eradication threshold for all treatments but the anti-toxin. It does not significantly affect the evolutionarily stable virulence. Second, we used a step function for the coverage, which is relaxed in electronic supplementary material, appendix S.6, where we use a sinusoidal function to capture a softer change from no coverage to full coverage. Qualitatively, our results and the associated public health recommendations do not depend of the shape on the periodic coverage function. Third, as commonly assumed in the literature [2,7], we used a density-independent birth rate for simplicity. However, the potential feedback between environmental fluctuations and density-dependent reproduction could be biologically relevant and worth a detailed investigation. Fourth, in contrast with the original model of Gandon *et al.* [7,8], we neglect here the possibility of multiple infections. With superinfection, anti-infection and anti-transmission

treatments actually select for lower virulence with constant coverage [7]. Although it is beyond the scope of this study, it would be interesting to see how the interplay between within-host selection and population-level environmental fluctuations could alter the selective pressures on virulence.

Importantly, our evolutionary analysis is based on an adaptive dynamics approach, which uncouples epidemiological and evolutionary timescales. Evolution is supposed to be much slower than epidemiological processes, due to rare mutations. However, many pathogens have high mutation rates and the dominant strain during an epidemic can differ from the strain that is selected in the long run. As experimentally demonstrated in [49], during an epidemic susceptible hosts are abundant and virulent strains investing in transmission are mostly selected, while less virulent strains are favoured at endemic equilibrium, where the proportion of susceptible is lower. It would be interesting to extend our model to take into account potential short-term evolutionary dynamics. Using quantitative genetics methods could help to shed light on these processes [50,51].

Finally, it would be interesting to test our theoretical predictions using field or experimental data. Unfortunately, the kind of field data needed to test our predictions require long-term studies of joint epidemiological and evolutionary dynamics, which are only beginning to appear. Nevertheless, our conclusions could be tested experimentally in microbial, or agricultural systems. Bacteria–phage interactions are well suited to explore the interplay between ecology and evolution in heterogeneous host–parasite interactions [49,52]. By periodically varying the influx of naive or treated susceptible bacteria and monitoring the effects on parasite prevalence and the evolution of phage virulence, it would be possible to test our predictions. We think our theoretical results provide an interesting foundation to guide experimental and empirical studies, which can potentially lead to useful recommendations to control and reduce the damages caused by infectious diseases.

Data accessibility. Code and data of numerical simulations are available from the Dryad Digital Repository: https://doi.org/10.5061/dryad.nzs7h44qx [53].

Authors' contributions. A.W. analysed the model, produced numerical analyses and figures and wrote the first draft of the manuscript. Both A.W. and S.L. contributed to subsequent versions of the manuscript. S.L. derived the analysis in electronic supplementary material, appendix S.4.3.

Competing interests. We declare we have no competing interests.

Funding. This work was funded by Agence Nationale de la Recherche grant ANR-16-CE35-0012-01 "STEEP" to S.L.

Acknowledgements. We thank Philippe Carmona and Sylvain Gandon for very useful discussions and suggestions. We are grateful to three anonymous reviewers for helpful comments.

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
