## [Peer Review File · Proceedings of the Royal Society B: Biological Sciences]

Review History

RSPB-2020-3007.R0 (Original submission)

Review form: Reviewer 1

Recommendation

Accept with minor revision (please list in comments)

Scientific importance: Is the manuscript an original and important contribution to its field?

Excellent

General interest: Is the paper of sufficient general interest?

Excellent

Quality of the paper: Is the overall quality of the paper suitable?

Excellent

Is the length of the paper justified?

Yes

Should the paper be seen by a specialist statistical reviewer?

No

Do you have any concerns about statistical analyses in this paper? If so, please specify them explicitly in your report.

No

It is a condition of publication that authors make their supporting data, code and materials available - either as supplementary material or hosted in an external repository. Please rate, if applicable, the supporting data on the following criteria.

Is it accessible?

Yes

Is it clear?

Yes

Is it adequate?

Yes

Do you have any ethical concerns with this paper?

No

Comments to the Author

This paper presents a theoretical model that analyzes the impact of fluctuating treatment environments on pathogen prevalence and virulence evolution. This model extends previous theoretical work on pathogen evolution in heterogeneous host environments by allowing treatment coverage to vary in time, which fills an important knowledge gap. The paper shows the potential benefit of periodic treatments for both managing virulence evolution and disease spread, and provides a framework for thinking about what types of treatments would most benefit from periodic usage. This paper is clearly and thoughtfully written, and has important implications for pathogen control. Mathematical details of the model and supporting code are available and transparent. My comments are all minor.

Minor comments

1. It might be helpful to have a legend for Figure 3 identifying the colors, rather than having this information in the text.
2. Figure 4 a & b are missing y-axis labels.
3. For Figure 5, descriptions of the colors used are missing the figure legend.

Review form: Reviewer 2

Recommendation

Accept with minor revision (please list in comments)

Scientific importance: Is the manuscript an original and important contribution to its field?

Excellent

General interest: Is the paper of sufficient general interest?

Excellent

Quality of the paper: Is the overall quality of the paper suitable?

Excellent

Is the length of the paper justified?

Yes

Should the paper be seen by a specialist statistical reviewer?

Yes

Do you have any concerns about statistical analyses in this paper? If so, please specify them explicitly in your report.

No

It is a condition of publication that authors make their supporting data, code and materials available - either as supplementary material or hosted in an external repository. Please rate, if applicable, the supporting data on the following criteria.

Is it accessible?

Yes

Is it clear?

Yes

Is it adequate?

No

Do you have any ethical concerns with this paper?

No

Comments to the Author

In this manuscript, the authors extend previous theory to examine how variation in treatment coverage and/or efficacy introduces heterogeneity in host populations and subsequently affects epidemiological and evolutionary dynamics.

Review synopsis: This paper is tackling an interesting question, as there is a real need for mathematical theory that examines how both intervention strategies and host heterogeneity influences disease invasion and pathogen evolution. The way the authors integrate these two factors is quite elegant and clever (especially considering that most studies to date on host heterogeneity tend to focus on genetic/genotypic heterogeneity). In general, I really appreciate the theoretical motivation and the goal of using analytical approaches to tackle such a complex question. The paper is overall well written, thoughtfully constructed, and addresses important questions.

I provide a few comments/suggestions. I hope that they will prove constructive.

My main comment/suggestion is that it would behoove the authors to explain how their approach differs from previous approaches examining leaky vaccines. More specifically, the methods and mathematical analyses developed here are quite sophisticated. Could the authors explain how this approach improves other, arguably more simplistic approaches? For example, Gandon et al., 2001, also examine how variation in the coverage and efficacy of anti-infection, anti-growth, anti-transmission, anti-toxin, and combination vaccines influence prevalence and the evolution of virulence. In their models, they assume equilibrium conditions, which is one major deviation from the model developed here (and potential contribution of the current study). This study is referenced throughout the manuscript, but I believe what would be beneficial is an explanation of when, where, and why the results converged or diverged (i.e., beyond the brief comparison in the Discussion). Such a discussion would help clarify the biological relevance of your study and highlight areas where it may/may not be particularly relevant; all of which would (should) guide future empirical studies aimed at integrating mathematical models with data.

Beyond that, I have only very minor suggestions/questions, which again, speaks to how well the manuscript is presented.

1. Consider explaining the biological interpretation/relevance of Equ. 12 a bit more – this was a bit hard to follow (even after going to the Supplementary material).
2. Figure S.4. an should be and...an the proportion of treatments...
3. Lines 277-279: Are there any studies that examine how environmental fluctuations influence pathogen evolution? This sentence makes it seem like there are none. If there are previous studies on this topic, this could be another useful comparison for your Discussion (I am harping on the Discussion not because it is not well written but because it primarily focuses on future prospects but gives very little coverage to highlighting the similarities/differences between your current study and others. This may be a matter of opinion, however, I suggest it because such comparisons can help propel the field by uncovering key missing biology or key differences to test empirically).

Review form: Reviewer 3

Recommendation

Accept with minor revision (please list in comments)

Scientific importance: Is the manuscript an original and important contribution to its field?

Excellent

General interest: Is the paper of sufficient general interest?

Excellent

Quality of the paper: Is the overall quality of the paper suitable?

Excellent

Is the length of the paper justified?

Yes

Should the paper be seen by a specialist statistical reviewer?

No

Do you have any concerns about statistical analyses in this paper? If so, please specify them explicitly in your report.

No

It is a condition of publication that authors make their supporting data, code and materials available - either as supplementary material or hosted in an external repository. Please rate, if applicable, the supporting data on the following criteria.

Is it accessible?

N/A

Is it clear?

N/A

Is it adequate?

N/A

Do you have any ethical concerns with this paper?

Yes

Comments to the Author

The manuscript uses mathematical models to examine the impact of a periodic vaccination regime on both the epidemiological dynamics of disease (assuming a simple SI framework) and on parasite evolution (using an adaptive dynamics framework). Four possible impacts of vaccination are considered. They show that most (but not all) treatments lead to reduced prevalence as coverage is increased. They show that two of the treatments lead to no selection on the parasite, but anti-growth and anti-toxin treatments select for increased virulence. In all treatments, periodic treatments could be seen as preferable to constant treatments.

I very much enjoyed reading this paper. It is on a topic of very general interest - the benefits and consequences of vaccination regimes - and on a specific issue that I think is really interesting - how might periodic treatment regimes select on the parasite. It is well written, the results are all well justified and explained and set against existing literature quite nicely.

I only have some minor comments that occurred to me while reading.

P2 L36 - Can you specify what you mean by the 'bad' habitat?

P3 L65 - Perhaps this is petty of me, but I wonder if the picture of a stick human should be replaced by something else. The model you propose would be really be more suited to a simple ecological setting.

P3 L77 - Sort of linked to the above, how reasonable is it to assume a constant birth rate with no density dependence? Does the population here stay steady enough in its dynamics for that to be reasonable?

P5 L111 - This is true in standard models, but certain models - particularly those with partial vaccination - mean $R_0 < 1$ is not sufficient for the disease to die out. Unless you are specifically assuming the case of an initially almost disease-free population?

P8 L147 - It might be good to include a sentence explaining why it is prevalence varies with period when the invasion threshold does not.

P8 L147 - Any idea why we get this non-linear shaped response? It implies there is a sweet spot of around 10 yearly(?) intervals to run your treatment program to reduce the prevalence.

P8 Figure 3 - As a colourblind reader I do struggle a little to tell your lines apart - especially red v green and blue v purple. Just making them thicker would be a help.

P9 L181 - Do you know for certain that the period of the dynamics is always the period of the vaccination? That is there are no period-doublings, etc? That would impact how you calculate your fitness.

P10 L184 - If this is the basic reproduction number could we not still call it R_0 ?

P13 Figure 4 - Please add y-axis labels to plots a and b.

P14 L261 - I wonder if this is slightly overstated. The methods have already been used by others haven't they? So while you are right that you have provided a very nice demonstration of it I don't know it is true that your method is 'new' in that sense is it?

P15 L279 - Again, maybe I'm missing something, but is your fundamental approach 'new' compared to say, references 40-42? Similarly a couple of studies by Ferris & Best do the same for host evolution - isn't this fundamentally the same approach of using Floquet theory to derive a fitness?

Decision letter (RSPB-2020-3007.R0)

19-Jan-2021

Dear Mrs Walter:

Your manuscript has now been peer reviewed and the reviews have been assessed by an Associate Editor. The reviewers' comments (not including confidential comments to the Editor) and the comments from the Associate Editor are included at the end of this email for your reference. As you will see, the reviewers and the Editors have raised some concerns with your manuscript and we would like to invite you to revise your manuscript to address them.

Research ethics:

Use of animals and field studies:

It is a condition of publication that you make available the data and research materials supporting the results in the article. Please see our Data Sharing Policies (<https://royalsociety.org/journals/authors/author-guidelines/#data>). Datasets should be deposited in an appropriate publicly available repository and details of the associated accession number, link or DOI to the datasets must be included in the Data Accessibility section of the article (<https://royalsociety.org/journals/ethics-policies/data-sharing-mining/>). Reference(s) to datasets should also be included in the reference list of the article with DOIs (where available).

Please submit a copy of your revised paper within three weeks. If we do not hear from you within this time your manuscript will be rejected. If you are unable to meet this deadline please let us know as soon as possible, as we may be able to grant a short extension.

Best wishes,
Professor Gary Carvalho
mailto:proceedingsb@royalsociety.org

Associate Editor

Board Member: 1

Comments to Author:

Thank you for submitting your manuscript "Epidemiological and evolutionary consequences of periodicity in treatment coverage" to Proceedings B. I have now received three expert reviews and evaluated the manuscript myself. We all see a lot of merit in this study and manuscript. Please take into account the suggestions from the reviewers in considering a revision of the submitted manuscript. Please also note suggestion from reviewer concerning how to make code more readily available.

Reviewer(s)' Comments to Author:

Referee: 1

Comments to the Author(s)

This paper presents a theoretical model that analyzes the impact of fluctuating treatment environments on pathogen prevalence and virulence evolution. This model extends previous theoretical work on pathogen evolution in heterogeneous host environments by allowing treatment coverage to vary in time, which fills an important knowledge gap. The paper shows the potential benefit of periodic treatments for both managing virulence evolution and disease spread, and provides a framework for thinking about what types of treatments would most benefit from periodic usage. This paper is clearly and thoughtfully written, and has important implications for pathogen control. Mathematical details of the model and supporting code are available and transparent. My comments are all minor.

Minor comments

1. It might be helpful to have a legend for Figure 3 identifying the colors, rather than having this information in the text.
2. Figure 4 a & b are missing y-axis labels.

3. For Figure 5, descriptions of the colors used are missing the figure legend.

Referee: 2

Comments to the Author(s)

In this manuscript, the authors extend previous theory to examine how variation in treatment coverage and/or efficacy introduces heterogeneity in host populations and subsequently affects epidemiological and evolutionary dynamics.

Review synopsis: This paper is tackling an interesting question, as there is a real need for mathematical theory that examines how both intervention strategies and host heterogeneity influences disease invasion and pathogen evolution. The way the authors integrate these two factors is quite elegant and clever (especially considering that most studies to date on host heterogeneity tend to focus on genetic/genotypic heterogeneity). In general, I really appreciate the theoretical motivation and the goal of using analytical approaches to tackle such a complex question. The paper is overall well written, thoughtfully constructed, and addresses important questions.

I provide a few comments/suggestions. I hope that they will prove constructive.

My main comment/suggestion is that it would behoove the authors to explain how their approach differs from previous approaches examining leaky vaccines. More specifically, the methods and mathematical analyses developed here are quite sophisticated. Could the authors explain how this approach improves other, arguably more simplistic approaches? For example, Gandon et al., 2001, also examine how variation in the coverage and efficacy of anti-infection, anti-growth, anti-transmission, anti-toxin, and combination vaccines influence prevalence and the evolution of virulence. In their models, they assume equilibrium conditions, which is one major deviation from the model developed here (and potential contribution of the current study). This study is referenced throughout the manuscript, but I believe what would be beneficial is an explanation of when, where, and why the results converged or diverged (i.e., beyond the brief comparison in the Discussion). Such a discussion would help clarify the biological relevance of your study and highlight areas where it may/may not be particularly relevant; all of which would (should) guide future empirical studies aimed at integrating mathematical models with data.

Beyond that, I have only very minor suggestions/questions, which again, speaks to how well the manuscript is presented.

1. Consider explaining the biological interpretation/relevance of Equ. 12 a bit more – this was a bit hard to follow (even after going to the Supplementary material).
2. Figure S.4. an should be and...an the proportion of treatments...
3. Lines 277-279: Are there any studies that examine how environmental fluctuations influence pathogen evolution? This sentence makes it seem like there are none. If there are previous studies on this topic, this could be another useful comparison for your Discussion (I am harping on the Discussion not because it is not well written but because it primarily focuses on future prospects but gives very little coverage to highlighting the similarities/differences between your current study and others. This may be a matter of opinion, however, I suggest it because such comparisons can help propel the field by uncovering key missing biology or key differences to test empirically).

Referee: 3

Comments to the Author(s)

The manuscript uses mathematical models to examine the impact of a periodic vaccination regime on both the epidemiological dynamics of disease (assuming a simple SI framework) and on parasite evolution (using an adaptive dynamics framework). Four possible impacts of vaccination are considered. They show that most (but not all) treatments lead to reduced

prevalence as coverage is increased. They show that two of the treatments lead to no selection on the parasite, but anti-growth and anti-toxin treatments select for increased virulence. In all treatments, periodic treatments could be seen as preferable to constant treatments.

I very much enjoyed reading this paper. It is on a topic of very general interest - the benefits and consequences of vaccination regimes - and on a specific issue that I think is really interesting - how might periodic treatment regimes select on the parasite. It is well written, the results are all well justified and explained and set against existing literature quite nicely.

I only have some minor comments that occurred to me while reading.

P2 L36 - Can you specify what you mean by the 'bad' habitat?

P3 L65 - Perhaps this is petty of me, but I wonder if the picture of a stick human should be replaced by something else. The model you propose would be really be more suited to a simple ecological setting.

P3 L77 - Sort of linked to the above, how reasonable is it to assume a constant birth rate with no density dependence? Does the population here stay steady enough in its dynamics for that to be reasonable?

P5 L111 - This is true in standard models, but certain models - particularly those with partial vaccination - mean $R_0 < 1$ is not sufficient for the disease to die out. Unless you are specifically assuming the case of an initially almost disease-free population?

P8 L147 - It might be good to include a sentence explaining why it is prevalence varies with period when the invasion threshold does not.

P8 L147 - Any idea why we get this non-linear shaped response? It implies there is a sweet spot of around 10 yearly(?) intervals to run your treatment program to reduce the prevalence.

P8 Figure 3 - As a colourblind reader I do struggle a little to tell your lines apart - especially red v green and blue v purple. Just making them thicker would be a help.

P9 L181 - Do you know for certain that the period of the dynamics is always the period of the vaccination? That is there are no period-doublings, etc? That would impact how you calculate your fitness.

P10 L184 - If this is the basic reproduction number could we not still call it R_0 ?

P13 Figure 4 - Please add y-axis labels to plots a and b.

P14 L261 - I wonder if this is slightly overstated. The methods have already been used by others haven't they? So while you are right that you have provided a very nice demonstration of it I don't know it is true that your method is 'new' in that sense is it?

P15 L279 - Again, maybe I'm missing something, but is your fundamental approach 'new' compared to say, references 40-42? Similarly a couple of studies by Ferris & Best do the same for host evolution - isn't this fundamentally the same approach of using Floquet theory to derive a fitness?

Author's Response to Decision Letter for (RSPB-2020-3007.R0)

See Appendix A.

Decision letter (RSPB-2020-3007.R1)

15-Feb-2021

Dear Mrs Walter

I am pleased to inform you that your manuscript entitled "Epidemiological and evolutionary consequences of periodicity in treatment coverage" has been accepted for publication in Proceedings B.

Open Access

Your article has been estimated as being 9 pages long. Our Production Office will be able to confirm the exact length at proof stage.

Paper charges

Sincerely,

Professor Gary Carvalho

Associate Editor:

Board Member

Comments to Author:

I commend the authors for having completed a thorough revision of their manuscript in response to each of the points raised by the reviewers.

Appendix A

Response to the decision letter

Associate Editor

Thank you for submitting your manuscript “Epidemiological and evolutionary consequences of periodicity in treatment coverage” to Proceedings B. I have now received three expert reviews and evaluated the manuscript myself. We all see a lot of merit in this study and manuscript. Please take into account the suggestions from the reviewers in considering a revision of the submitted manuscript. Please also note suggestion from reviewer concerning how to make code more readily available.

We thank the editor and reviewers for their positive appreciation of our work and their very helpful comments. We have taken all these comments into account when preparing our revised manuscript. All code needed to reproduce the figures in the article has been uploaded with our submission and will be made available online.

In the revised version of the manuscript, the main changes to the text are indicated in bold, sans-serif font.

With our best wishes,

Alicia Walter and Sébastien Lion

Code and data of numerical simulations can be found in Dryad at :

<https://datadryad.org/stash/share/vQCOuhLSQii9HfqTOhmfmPwOFh9zwV2jWGEy5lexfjA>

Referee : 1

This paper presents a theoretical model that analyzes the impact of fluctuating treatment environments on pathogen prevalence and virulence evolution. This model extends previous theoretical work on pathogen evolution in heterogenous host environments by allowing treatment coverage to vary in time, which fills an important knowledge gap. The paper shows the potential benefit of periodic treatments for both managing virulence evolution and disease spread, and provides a framework for thinking about what types of treatments would most benefit from periodic usage. This paper is clearly and thoughtfully written, and has important implications for pathogen control. Mathematical details of the model and supporting code are available and transparent. My comments are all minor.

We thank the reviewer for their positive appreciation and helpful comments.

Minor comments

1. It might be helpful to have a legend for Figure 3 identifying the colors, rather than having this information in the text.

Thank you for this suggestion. We have made the suggested change.

2. Figure 4 a & b are missing y-axis labels.

This has been fixed.

3. For Figure 5, descriptions of the colors used are missing the figure legend.

This has been fixed.

Referee : 2

In this manuscript, the authors extend previous theory to examine how variation in treatment coverage and/or efficacy introduces heterogeneity in host populations and subsequently affects epidemiological and evolutionary dynamics.

Review synopsis: This paper is tackling an interesting question, as there is a real need for mathematical theory that examines how both intervention strategies and host heterogeneity influences disease invasion and pathogen evolution. The way the authors integrate these two factors is quite elegant and clever (especially considering that most studies to date on host heterogeneity tend to focus on genetic/genotypic heterogeneity). In general, I really appreciate the theoretical motivation and the goal of using analytical approaches to tackle such a complex question. The paper is overall well written, thoughtfully constructed, and addresses important questions.

I provide a few comments/suggestions. I hope that they will prove constructive.

We thank the reviewer for their positive appreciation and very constructive comments.

My main comment/suggestion is that it would behoove the authors to explain how their approach differs from previous approaches examining leaky vaccines. More specifically, the methods and mathematical analyses developed here are quite sophisticated. Could the authors explain how this approach improves other, arguably more simplistic approaches ? For example, Gandon et al., 2001, also examine how variation in the coverage and efficacy of anti-infection, anti-growth, anti-transmission, anti-toxin, and combination vaccines influence prevalence and the evolution of virulence. In their models, they assume equilibrium conditions, which is one major deviation from the model developed here (and potential contribution of the current study). This study is referenced throughout the manuscript, but I believe what would be beneficial is an explanation of when, where, and why the results converged or diverged (i.e., beyond the brief comparison in the Discussion). Such a discussion would help clarify the biological relevance of your study and highlight areas where it may/may not be particularly relevant ; all of which would (should) guide future empirical studies aimed at integrating mathematical models with data.

This is a very helpful suggestion, and we have tried to include further discussion of the seminal work by Gandon et al at the end of section 4.1 and 4.2, and in the discussion.

Although an extensive discussion of the results by Gandon et al is impossible, the reviewer's remark has led us to realise that a specific result of the model by Gandon et al could seem at odds with our own results. That is, for anti-infection and anti-transmission vaccines, Gandon et al predicted a negative relationship between ESS virulence and vaccine efficacy, whereas we predict a flat line. This discrepancy is due to an additional assumption of Gandon et al, which is the possibility of superinfection. In our model, we neglect superinfection because of the added complications caused

by the interplay between within-host selection and environmental fluctuations in coverage. We now discuss this point in more detail in our manuscript.

1. Consider explaining the biological interpretation/relevance of Eq. 12 a bit more – this was a bit hard to follow (even after going to the Supplementary material).

We have clarified this section.

2. Figure S.4. an should be and...an the proportion of treatments...

This has been fixed.

3. Lines 277-279 : Are there any studies that examine how environmental fluctuations influence pathogen evolution? This sentence makes it seem like there are none. If there are previous studies on this topic, this could be another useful comparison for your Discussion (I am harping on the Discussion not because it is not well written but because it primarily focuses on future prospects but gives very little coverage to highlighting the similarities/differences between your current study and others. This may be a matter of opinion, however, I suggest it because such comparisons can help propel the field by uncovering key missing biology or key differences to test empirically).

We have clarified this paragraph. Although ours is not the first study to analyse pathogen evolution in fluctuating environments, many look at unstructured populations (i.e. only one equation is needed to describe the mutant dynamics) and the few studies that look at structured populations (notably the papers by Ferris & Best, to which we add a reference) have to resort to numerical explorations. In contrast, at least to our knowledge, our study is the first to derive an analytical expression of the selection gradient. In addition this expression depends on a dynamical concept of reproductive value, which allows for a direct comparison between the selection gradients in constant vs periodic environments, and also provides additional biological insight.

Referee : 3

The manuscript uses mathematical models to examine the impact of a periodic vaccination regime on both the epidemiological dynamics of disease (assuming a simple SI framework) and on parasite evolution (using an adaptive dynamics framework). Four possible impacts of vaccination are considered. They show that most (but not all) treatments lead to reduced prevalence as coverage is increased. They show that two of the treatments lead to no selection on the parasite, but anti-growth and anti-toxin treatments select for increased virulence. In all treatments, periodic treatments could be seen as preferable to constant treatments.

I very much enjoyed reading this paper. It is on a topic of very general interest - the benefits and consequences of vaccination regimes - and on a specific issue that I think is really interesting - how might periodic treatment regimes select on the parasite. It is well written, the results are all well justified and explained and set against existing literature quite nicely.

I only have some minor comments that occurred to me while reading.

We thank the reviewer for their positive appreciation and their detailed comments, which have helped us to clarify the main text of the manuscript.

P2 L36 - Can you specify what you mean by the 'bad' habitat ?

We have added a sentence to clarify this point. We mean that treated hosts represent bad habitats from the point of view of the parasite, because a pathogen will have reduced fitness in a treated host compared to an untreated host. Of course, from a public health perspective, a bad habitat for the parasite is a good (treated) host.

P3 L65 - Perhaps this is petty of me, but I wonder if the picture of a stick human should be replaced by something else. The model you propose would be really be more suited to a simple ecological setting.

We understand the reviewer's concern, and we have carefully thought about what to do. In the end, we have decided to keep the figure as it is, and to add a clarifying sentence in the caption, noting that the human host is only used for illustration purposes. We made this choice out of simplicity, and because we think that, because our SI model is first and foremost conceptual, any specific choice of host could be criticised as unrealistic. However, the simplicity of a model is not always predictive of its accuracy (think of how SIR-like models can work well in human populations), so, although we agree there are many ways to improve the realism of our model, we think our presentation makes it sufficiently clear that our aim is to analyse a general rather than a specific biological scenario.

P3 L77 - Sort of linked to the above, how reasonable is it to assume a constant birth rate with no density dependence? Does the population here stay steady enough in its dynamics for that to be reasonable?

Our assumption of a density-independent birth rate was made for two main reasons. First, it is commonly made in the literature (see e.g. Keeling & Rohani, 2008, chapter 5) and this allows us to directly compare our results with those of Gandon et al 2001 without adding any additional effect of density-dependence. Second, although reproduction is of course a density-dependent process, our assumption is the simplest that allows the host population to be regulated in the absence of pathogen. In a model with constant coverage, the equilibrium density of hosts is simply $S^* = d/b$, whereas, with density-dependent growth (i.e. $dS/dt = bS - dS$), we would have exponential growth in the absence of the disease and the pathogen would be the only factor that regulates the host population, which we don't find realistic. Alternatively, we could use a logistic growth assumption (i.e. $dS/dt = bS(1-N) - dS$) but this leads to more complex dynamics and makes the comparison with the seminal results by Gandon et al 2001 more difficult.

However, we agree with the reviewer that it would be interesting to investigate in more detail the interplay between density-dependent reproduction and fluctuations in coverage, and we have added a sentence in the discussion to clarify this point.

P5 L111 - This is true in standard models, but certain models - particularly those with partial vaccination - mean $R_0 < 1$ is not sufficient for the disease to die out. Unless you are specifically assuming the case of an initially almost disease-free population?

The reviewer is correct that we define R_0 in an initially almost disease-free population. We have clarified the sentence.

P8 L147 - It might be good to include a sentence explaining why it is prevalence varies with period when the invasion threshold does not.

We have added a sentence to that effect at the end of section 3. The answer is that, for our model, the disease-free state is weakly dependent on the period, whereas epidemiological feedbacks due to disease transmission make the endemic attractor much more sensitive to the forcing period. As a result, R_0 only slightly depends on the period (or can even be independent of T , as for r_1 and r_3 treatments, due to symmetries in the model).

P8 L147 - Any idea why we get this non-linear shaped response? It implies there is a sweet spot of around 10 yearly (?) intervals to run your treatment program to reduce the prevalence.

It is expected that as T becomes large, the response should saturate as the system tends towards an alternation of homogeneous equilibria with our choice of periodic function for the treatment. So we expect some non-linearity, but we do not have a clear explanation for the specific shape of the response for small and intermediate periods.

P8 Figure 3 - As a colourblind reader I do struggle a little to tell your lines apart - especially red v green and blue v purple. Just making them thicker would be a help.

We thank the reviewer for this very important suggestion. We have made two main changes to the figures. First, we have added labels so that it is immediately clear which type of treatment the curve represents. Second, we have modified our colour palette to use a palette adapted for colour blindness and proposed by Wong, Nature, 2011. We have tested this using various colourblindness filters and hope the end result is more readable.

P9 L181 - Do you know for certain that the period of the dynamics is always the period of the vaccination? That is there are no period-doublings, etc? That would impact how you calculate your fitness.

This is a very good point. Although we have no mathematical proof, extensive numerical simulations suggest that the period of the resident attractor equals the period of the treatment and that there is no period doubling. We have added a sentence to clarify this point.

Of course, this could be different for other systems, and the period on the resident attractor could even be a function $\tau(zw)$ of the resident trait, in which case the appropriate period on which to integrate would need to be $\tau(zw)$.

P10 L184 - If this is the basic reproduction number could we not still call it R_0 ?

It is true that both R and R_0 are conceptually both basic reproduction numbers. However, they are evaluated in different environments (either the resident endemic population or the disease-free population), and depend on a different number of variables (either the resident and mutant traits, or only the resident trait), so we think it is less confusing to distinguish the two. Furthermore, for the anti-infection and anti-transmission vaccines, R_0 et R have almost the same expression, which may add to the confusion for some readers. We follow therefore previous practice (see e.g. Lion & Metz 2018) and use different symbols for the two quantities.

P13 Figure 4 - Please add y-axis labels to plots a and b.

This has been fixed.

P14 L261 - I wonder if this is slightly overstated. The methods have already been used by others haven't they ? So while you are right that you have provided a very nice demonstration of it I don't know it is true that your method is 'new' in that sense is it ?

While it is true that we are not the first to use Floquet exponents to calculate invasion fitness in periodic environments (see e.g. the studies by Donnelly et al and Ferris & Best that we cite), our approach is the first to lead to a biologically meaningful analytical expression of the selection gradient in class-structured populations. The studies of Ferris & Best give only numerical calculations of the Floquet exponent when the mutant dynamics need to be described by a matrix (i.e. when recovery is non-zero in their model), whereas our approach allows us to calculate a first-order approximation of the Floquet exponent which depends on the reproductive values of each class on the resident periodic attractor. This is the first application of the dynamical extension of reproductive values proposed by Lion (2018, Am Nat), and a methodological paper on this approach in periodic environments is in preparation (Lion & Gandon, in prep).

We have clarified this point in the discussion.

P15 L279 - Again, maybe I'm missing something, but is your fundamental approach 'new' compared to say, references 40-42? Similarly a couple of studies by Ferris & Best do the same for host evolution - isn't this fundamentally the same approach of using Floquet theory to derive a fitness?

We have added some references to the studies by Ferris & Best, and have highlighted the key differences between their approach and ours.